



# Snowball Earth transitions from Last Glacial Maximum conditions provide an independent upper limit on Earth's climate sensitivity

Martin Renoult[1,2], Navjit Sagoo[2], Johannes Hörner[3], and Thorsten Mauritsen[2]

[1]Department of Geological Sciences, Bolin Centre for Climate Research, Stockholm University, Stockholm, Sweden

[2]Department of Meteorology, Bolin Centre for Climate Research, Stockholm University, Stockholm, Sweden

[3]Department of Meteorology and Geophysics, University of Vienna, Vienna, Austria

**Correspondence:** Martin Renoult (martin.renoult@misu.su.se)

**Abstract.** Geological evidence of a snowball Earth state indicate persistent tropical sea ice cover during the Neoproterozoic (>635 million years ago). Current theory is that a strengthening of the positive surface albedo feedback with cooling temperatures, eventually exceeding the sum of all other feedbacks, leads to a global climate instability. Several recent high sensitivity climate models with strongly positive cloud feedbacks have not been able to simulate the much warmer Last Glacial Maximum
state, suggestive that they cool excessively in response to a modest decrease in atmospheric carbon dioxide levels and therefore enter the snowball instability by this mechanism. Using a coupled Earth system model, MPI-ESM1.2, we show that clouds accelerate the transition to a snowball Earth state and reduce the radiative forcing required to trigger the climate instability. Positive cloud feedbacks over tropical oceans and ahead of the sea-ice edge act to cool down the oceans and promote sea ice formation. Regardless, when approached slowly the snowball Earth transitions appear to occur around a global mean temper-
ature of zero degree Celsius, simultaneously with the sea ice edge advancing into the sub-tropics thereby strengthening the surface albedo feedback. This temperature threshold, if supported by several climate models, could be used as a novel and independent constraint on the upper bound of climate sensitivity. Currently, using the results from MPI-ESM1.2, we find it is implausible that Earth's climate sensitivity exceeds 5.5 K (4.4 – 6.6, 90 percent confidence).

## 1 Introduction

During the Neoproterozoic (>635 million years ago), geological evidences support the formation of persistent sea-ice within the tropical regions, referred as snowball Earth states. Because ice is highly reflective, the positive surface albedo feedback strengthens and exceeds the sum of other feedbacks while the Earth cools down. The evolution of climate feedbacks as a function of temperature is referred as state-dependency. Whilst climate models agree that the surface albedo feedback increases in cold climates, other climate feedbacks such as clouds are less understood in the context of cooling (e.g. Braun et al., 2022).



State-dependency in climate feedbacks has often been studied in warming climates (e.g. Caballero and Huber, 2013; Jonko et al., 2013; Meraner et al., 2013), but rarely in cold climates (e.g. Colman and McAvaney, 2009).

In this study, we quantify cloud feedbacks as the Earth transits towards a snowball Earth state from pre-industrial conditions under low $CO_2$ concentration. In particular, we focus on the contribution of cloud feedbacks over tropical oceans, as well as the interaction between cloud and sea-ice albedo feedbacks. Our motivation to investigate climate feedbacks in the snowball Earth

transition stems from the fact that a number of models of the paleoclimate modelling intercomparison project phase 4 (PMIP4) fail to simulate the Last Glacial Maximum (LGM), a cold period with large ice sheets 21 000 years ago (e.g. Kageyama et al., 2021), despite being able to simulate warm paleoclimates (e.g. Pliocene, 3.2 million years ago (Haywood et al., 2020), Eocene, 50 million years ago (Lunt et al., 2021)). These climate models start transiting to a snowball state at temperatures substantially cooler than indicated by LGM reconstructions (e.g. Zhu et al., 2021a). Since they have strong positive cloud

feedbacks they consequently exhibit high climate sensitivities, which is the long term temperature change after an abrupt doubling of atmospheric $CO_2$ from pre-industrial concentrations. It could be possible to use the models simulating stable LGM states to provide an independent upper limit constraint on climate sensitivity by estimating the temperature at which the Earth transits towards an unstable snowball Earth state. To approach this problem with other climate models, we conclude our study with a short and easy-to-replicate experimental design of modern snowball Earth simulations. The results of this experiment

are relevant for both climate sensitivity, as shown in this paper, as well as understanding the challenges around setting up the LGM simulation, a notoriously difficult paleoclimate to model.

## 2 Methods

We use the Max Planck Institute for Meteorology Earth system model version 1.2, MPI-ESM1.2 (Mauritsen et al., 2019) to simulate snowball Earth transitions from pre-industrial (PI) and LGM initial conditions. We perform abrupt and sustained

changes of atmospheric $CO_2$ concentrations or solar constant from the equilibrium states of PI or LGM. Continents, non-$CO_2$ greenhouse gases and orbital configurations are kept as in PI or LGM. PI simulations use the coarse resolution MPI-ESM1.2-CR (T31 spectral truncation, 31 atmospheric levels), which is faster and numerically more stable to extreme forcing. LGM boundary conditions are only available in the low resolution MPI-ESM1.2-LR (T63, 47 atmospheric levels). Radiative differences between the two models are small, and the climate sensitivity of MPI-ESM1.2-CR is slightly higher. While we do

not expect state-dependency to vary much across models, individual feedbacks are likely to slightly differ so we also initiate a few simulations from PI with MPI-ESM1.2-LR. Details of all simulations are summarised in Table 1.

The growth of thick sea-ice leads to numerical instabilities in the model (> 12 meters). We do not artificially limit sea-ice growth as in other studies (Voigt and Marotzke, 2010; Voigt et al., 2011; Voigt and Abbot, 2012), as it generates latent heat at the base of sea-ice (Marotzke and Botzet, 2007) which changes the required $CO_2$ forcing for snowball Earth initiation (Hörner

et al., 2022).




**Table 1.** Summary of the runs performed for this study. PI = Pre-industrial, LGM = Last Glacial Maximum. Solar constant is expressed in percentage of pre-industrial solar constant (1361 $Wm^{-2}$). *The runs started from an equilibrium state and ran for 100 years. **The runs were manually stopped and are expected to reach equilibrium in a cold non-snowball state.

| Run | Length (years | Solar constant (%) | $CO_2$ (ppm) | Boundaries | Comments |
|---|---|---|---|---|---|
| Pre-industrial | * | 100 | 284.32 | PI | MPI-ESM1.2-CR |
| LGM | * | 100 | 190 | LGM | MPI-ESM1.2-LR |
| 2xCO2 | 150 | 100 | 568.64 | PI | MPI-ESM1.2-CR |
| abrupt50ppm | 1156** | 100 | 50 | PI | MPI-ESM1.2-CR |
| 1/8xCO2 | 1142 | 100 | 35.54 | PI | MPI-ESM1.2-CR |
| 1/16xCO2 | 639 | 100 | 17.80 | PI | MPI-ESM1.2-CR |
| 1/32xCO2 | 442 | 100 | 8.90 | PI | MPI-ESM1.2-CR |
| 1/64xCO2 | 338 | 100 | 4.45 | PI | MPI-ESM1.2-CR |
| 1/128xCO2 | 280 | 100 | 2.22 | PI | MPI-ESM1.2-CR |
| 1/256xCO2 | 245 | 100 | 1.11 | PI | MPI-ESM1.2-CR |
| 1/512xCO2 | 219 | 100 | 0.56 | PI | MPI-ESM1.2-CR |
| 1/1024xCO2 | 230 | 100 | 0.28 | PI | MPI-ESM1.2-CR |
| 1/2056xCO2 | 194 | 100 | 0.14 | PI | MPI-ESM1.2-CR |
| 1/128xCO2-fixalb | 1046** | 100 | 2.22 | PI | locked albedo feedback |
| 1/128xCO2-fixq | 632** | 100 | 2.22 | PI | locked water vapor feedback |
| 1/128xCO2-fixcld | 946 | 100 | 2.22 | PI | locked cloud feedback |
| 1/512xCO2-fixcld | 518 | 100 | 0.56 | PI | locked cloud feedback |
| 1/256xCO2-fixcld | 648 | 100 | 1.11 | PI | locked cloud feedback |
| 1/64xCO2-fixcld | 1453** | 100 | 4.45 | PI | locked cloud feedback |
| 1/32xCO2-fixcld | 1312** | 100 | 8.90 | PI | locked cloud feedback |
| lgm-S94 | 290 | 94 | 190 | LGM | MPI-ESM1.2-LR |
| lgm-S93 | 198 | 93 | 190 | LGM | MPI-ESM1.2-LR |
| lgm-S92 | 179 | 92 | 190 | LGM | MPI-ESM1.2-LR |
| lgm-S91 | 148 | 91 | 190 | LGM | MPI-ESM1.2-LR |
| lgm-S89 | 113 | 89 | 190 | LGM | MPI-ESM1.2-LR |
| lgm-S85 | 75 | 85 | 190 | LGM | MPI-ESM1.2-LR |
| S92-LR | 179 | 92 | 284.32 | PI | MPI-ESM1.2-LR |
| S85-LR | 166 | 85 | 284.32 | PI | MPI-ESM1.2-LR |
| CESM1.2 | 210 | 100 | 2.22 | PI | CESM1.2 |

## 2.1 Climate feedbacks calculations

Climate feedbacks are diagnosed using the partial radiative perturbation (PRP) method (Wetherald and Manabe, 1988; Colman and McAvaney, 1997). Details of its online implementation in MPI-ESM1.2 are in Meraner et al. (2013). The PRP method calculates individual contributions of surface albedo, clouds, temperature and water vapor changes to top-of-atmosphere fluxes, by exchanging related variables between a control climate state and the transient state analysed. Because the length of each run varies with the forcing amplitude, we compute climate feedbacks by regressing the top-of-atmosphere radiation balance changes arising from albedo, clouds, temperature, water vapor changes over global temperature ranges of 5°C.

We furthermore perform runs where we separately lock surface albedo, clouds or water vapor in the radiation calculations to the control state, i.e. the corresponding feedbacks do not contribute to the radiation balance. The implementation in MPI-ESM1.2 is described in Mauritsen et al. (2013). These locked-feedback transient simulations read the pre-industrial control



albedo, clouds, temperature and humidity and impose them on the radiation parameterization regardless of the changes the system is experiencing, such as the increasing extent of sea-ice.

## 2.2 Constraint on Earth's climate sensitivity

The relationship between LGM simulated temperatures and the climate sensitivity of models have been used in emergent con-
straint framework to infer the Earth's true climate sensitivity owing to geological reconstructions of LGM temperatures (Hargreaves et al., 2012; Schmidt et al., 2014; Renoult et al., 2020, 2023). The novel constraint on Earth's climate sensitivity developed here uses the same relationship, where the regression parameters are calculated via ordinary least squares method, but we replace the geological reconstruction of the LGM by the temperature at which the Earth transits to a snowball Earth. This constraint relies on the simple fact that the LGM was a stable glacial climate, and therefore the highest value of Earth's
climate sensitivity before the LGM enters a snowball state represents the upper limit on climate sensitivity.

Recently, Renoult et al. (2023) showed that structural uncertainties and state-dependencies in climate feedbacks hinder the relationship between LGM temperature and climate sensitivity, which is computed from warm simulations (4 times pre-industrial $CO_2$ (Gregory et al., 2004)). We minimize the issue by adding to the PMIP4 ensemble an ensemble of CESM models (CESM1.2, CESM1.3 and CESM2.1) in a similar way as described by Renoult et al. (2023). Besides increasing the size of
the PMIP4 ensemble, single-model ensembles (or in this case, single-family models) have reduced structural uncertainties, in particular regarding ice sheet forcing, albeit they could add statistical dependency issues within the relationship. Here we assume the three CESM models are sufficiently different to avoid this issue.

## 3 Analysis of surface albedo and cloud feedbacks

We perform experiments with abrupt decreases of $CO_2$ concentrations ranging from 1/8 to 1/2048 of pre-industrial $CO_2$ concen-
tration using the MPI-ESM1.2-CR model to simulate snowball Earth inception and breakdown climate feedbacks (Section 2). For clarity, we report temperature anomalies to pre-industrial in units of K, and absolute temperatures in °C. The strengthening surface albedo feedback has often been considered the main driver in the snowball Earth instability, yet positive cloud feedbacks exceed the surface albedo feedback during the initial 10 K of cooling (Fig. 1). This strengthening arises from positive tropical cloud feedbacks (Fig. 2) due to an increase in shallow cloud coverage over the tropical oceans (Appendix Fig. B1). Be-
low −20 K relative to pre-industrial, cloud feedbacks switch sign and are globally negative and weaker than the surface albedo feedback, although there still exists strong positive cloud feedbacks ahead of the advancing sea-ice edge in both hemispheres (Fig. 2). Indeed, because the sea-ice surface is cold, clouds are preferentially over open water which is warmer and a source of moisture (e.g. Wall et al., 2017), where they facilitate further cooling of the ocean. All in all, cloud feedbacks substantially contribute to early cooling in response to reduced $CO_2$. During the transition, the positive surface albedo feedback exceeds
the combined cloud, temperature and water vapor feedbacks at temperatures between −15 K and −20 K below pre-industrial (Fig. 3). This leads to a global instability and is concomitant with the acceleration of southern sea-ice edge into the sub-tropics, resulting in strong global cooling towards the snowball state.





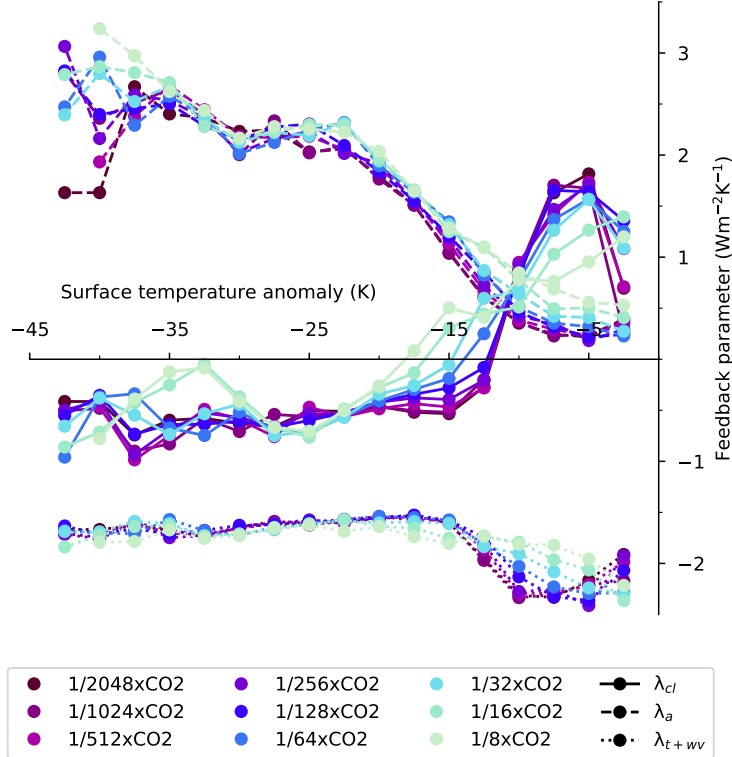

**Figure 1.** Evolution of the climate feedback parameters of clouds ($\lambda_{cl}$), albedo ($\lambda_a$) and temperature and water vapor combined ($\lambda_{t+wv}$) with global surface temperature anomaly compared to pre-industrial (K) under all $CO_2$ forcing.

The diagnostics presented above is suggestive that cloud feedback may be an important climate driver towards the snow ball transition. To demonstrate that this is indeed the case, we isolate the importance of each feedback using the feedback locking technique (Section 2) on various simulations, where each feedback mechanism is disabled one after the other (Fig. 3-B). When the cloud feedback is locked, the instability is reached at a similar global mean temperature as in the non-locked simulations, between $-15$ K and $-20$ K (Fig. 3-A). However, cloud feedback locking drastically increases the forcing required to initiate the snowball Earth inception, and considerably slow down the transition to snowball Earth. We find that simulations at $CO_2$ concentration of 1/64 times pre-industrial levels and lower are required to trigger the transition with locked cloud feedbacks, which is a factor ten less than the concentration required with interactive cloud feedbacks. The surface albedo-locked simulation does not reach a snowball instability as it enters quasi-equilibrium below the transition temperature of around $-15$ K. Similarly, the simulation reaches a quasi-equilibrium below $-10$ K when the water vapor feedback is locked (Appendix Fig. B2). The water vapor feedback actually does not strengthen with cooling so is not itself the cause of the snowball instability, however its large positive feedback is important for allowing the surface albedo to trigger the instability.



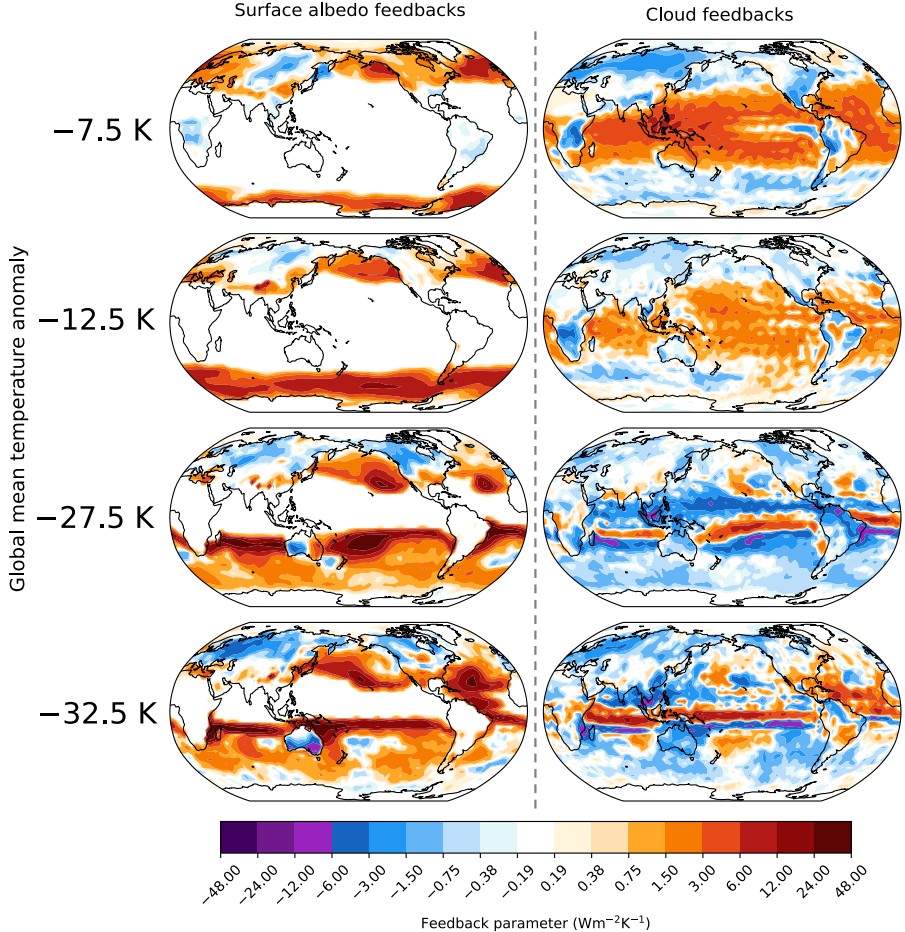

**Figure 2.** Maps of the surface albedo (left column) and cloud (right column) feedbacks at four average global temperatures in the 1/16xCO2 simulation.

Thus, the role of cloud feedback is different from that of the surface albedo feedback: 1) positive cloud feedbacks facilitate an early tropical cooling, 2) positive cloud feedbacks ahead of the advancing sea-ice edge and its effect on tropical oceans accelerate the transition to snowball Earth and decrease the temperature of instability, as summarized in Fig. 4; 3) cloud feedbacks substantially increase the threshold $CO_2$ level required for snowball Earth initiation and 4) the entry of sea-ice into the southern subtropics results in the surface albedo feedback exceeding the sum of the negative feedbacks. Whereas the albedo feedback is responsible for the transition and the main driver to the complete glaciation, our results emphasise an important contribution of cloud feedbacks. And as it happens, cloud feedbacks are also the main cause of inter model spread in climate sensitivity (e.g. Zelinka et al., 2020), a fact which we shall exploit in Section 5.



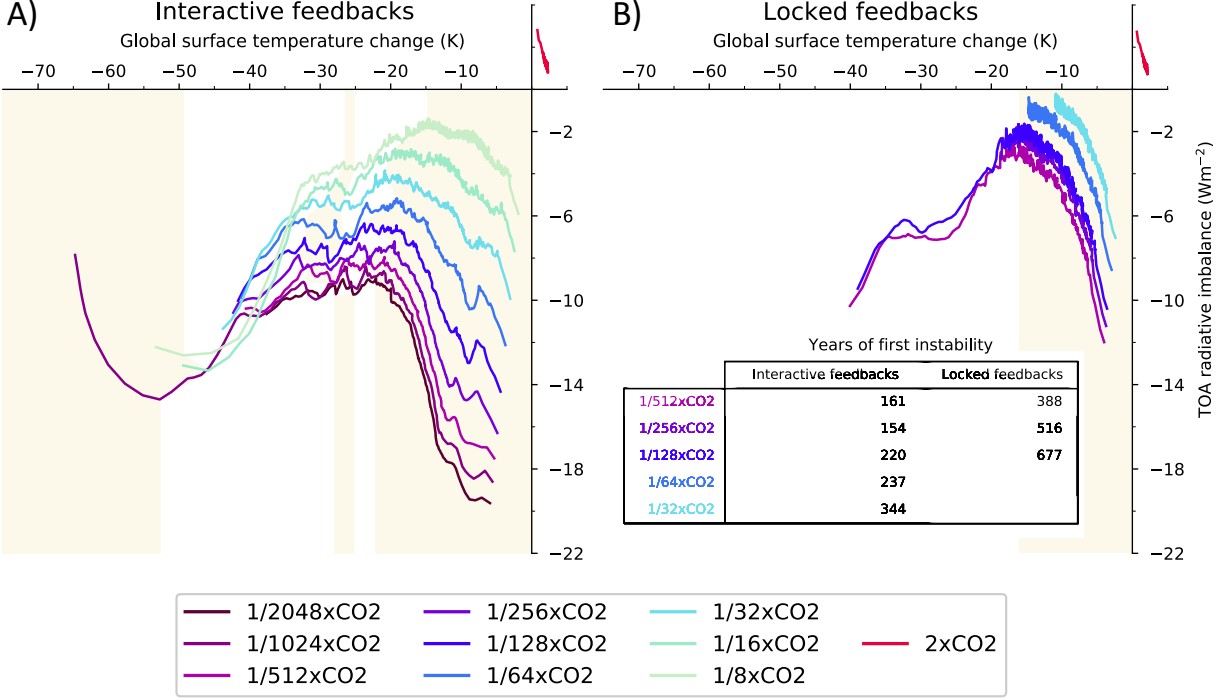

**Figure 3.** A) Top-of-atmosphere radiative imbalance ($\mathrm{Wm^{-2}}$) versus global surface temperature anomaly with pre-industrial conditions (K) under varying $CO_2$ forcing and B) same but with cloud feedbacks locked to that of the control state. A run with an abrupt doubling of $CO_2$ (2xCO2) is shown for comparison. Colored phases correspond to a stable climate (negative feedbacks), white phases are unstable climate (positive feedbacks). The years of the first instability between interactive and locked feedbacks are annotated.

## 4 Model evidence for the snowball transition temperature

The temperature at which the climate transits towards a snowball Earth state is broadly similar across the different $CO_2$ forcing
experiments (Fig. 3), as well as whether the experiment is initialized from a PI or an LGM state, as shown in Appendix A, which demonstrates the importance of background temperature control over the climate feedbacks which drive the transition. In particular, sea-ice formation is primarily temperature-dependent, and the amplitude of the sea-ice albedo feedback is practically independent of the strength of the negative forcing (Fig. 1). We can expect such state-dependency to be similar across climate models, which would contribute to them having a similar temperature of entering the snowball transition. Similarly, the winds
pushing sea-ice equatorwards (Appendix Fig. B3), in connection with temperature contrast between sea-ice and open ocean, and which are concomitant with positive cloud feedbacks ahead of sea-ice edge are also expected to behave in similar ways across models since they broadly simulate the same circulation. A simple geometric argument for a transition temperature close to 0°C is that it happens when the ice edge enters the sub-tropics at about 30 degrees latitude, leaving about half the Earth still



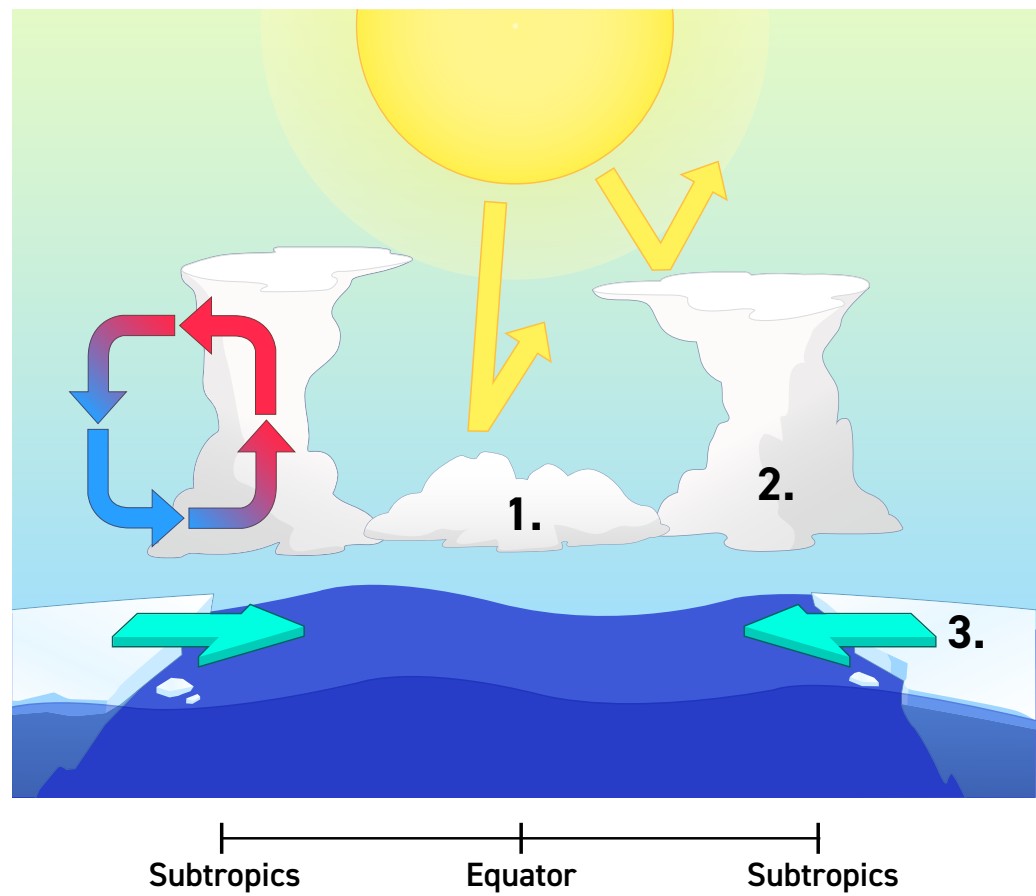

**Figure 4.** Contribution of cloud feedbacks to ocean cooling and interaction with sea-ice. 1. Tropical surface cooling by positive low-level cloud feedback. 2. Sea surface cooling ahead of ice edge. 3. Faster sea-ice advance, facilitated by winds pushing sea-ice equatorwards (Appendix Fig. B3).

ice free and hence above the freezing temperature, and the other half at temperatures below. When abruptly decreasing the

$CO_2$ concentration to 50 ppm (around 1/4 of pre-industrial $CO_2$), we find hints of instability near the global mean temperature of 0°C (−15 K relative to pre-industrial), but the model seems to be stable for 1848 years until it crashes due to thick sea-ice (Section 2). This indicates that it is unlikely that the inception temperature happens at a temperature substantially warmer than 0°C.

Nevertheless, the transition temperatures of each phase show a slight shift to lower values under stronger negative forcing.

Therefore, the climate system deviates from pure state-dependent behaviour as the strength of the radiative cooling and the





speed of transition to snowball Earth increases. These time-dependency effects have often been described in the literature following extreme cooling rates (Bendtsen and Bjerrum, 2002; Marotzke and Botzet, 2007), and we believe to be due to the large heat capacity of the ocean, as well as differences in ocean mixed layer depths and direct $CO_2$ forcing across different regions: in certain circumstances, this can result in local temperatures dropping faster than global temperatures (Appendix
Fig. B4-A), and hence out of sync with the sea-ice extent controlling the surface albedo feedback. Ocean convective mixing is also increasingly more efficient at evacuating heat for larger surface cooling, indicated by an increasing peak of the strength of the Atlantic meridional overturning circulation (AMOC) in Appendix Fig. B4-B. The associated ocean currents on the contrary drag sea-ice polewards and can slow down the sea-ice edge advance (Voigt and Abbot, 2012). All in all, we suggest slow, low forcing simulations are preferable when analysing snowball Earth transitions, as 1) fast transitions to snowball Earth are hardly
realistic, as geological snowball states may form over millions of years (e.g. Schrag et al., 2002) and 2) fast transitions involves temporal effects which would depart from state-dependency. All together, these considerations support the higher transition temperature close to 0°C found in the experiments with weak forcing.

## 5 Motivation for simulations and potential constraint on climate sensitivity

While we expect the state-dependency to behave similarly across models, it is currently not known whether the temperature
threshold of 0°C that we find in MPI-ESM1.2-CR is indeed universal. The CESM model family has notoriously performed several snowball Earth simulations (Yang et al., 2012b, a; Eisenman and Armour, 2024), and while the model CESM2 also transits towards a snowball Earth around 0°C under similar $CO_2$ forcing (Eisenman and Armour, 2024), models such as CCSM3 and CCSM4 can have stable waterbelt states with abrupt but smaller transitions. To further test this, we perform an abrupt 1/128xCO2 simulation with the model CESM1.2. CESM1.2 has been used for several deep-time simulations (Li et al.,
2022), but to our knowledge never for snowball Earth states. CESM1.2 shows a similar instability than MPI-ESM1.2 around a temperature of 0°C, considering the time-dependency effects discussed in this study (Fig. 5).

There is an ongoing discussion on whether climate models with large cloud feedbacks and consequently high climate sensitivity are compatible with the LGM (Zhu et al., 2022; Burls and Sagoo, 2022). Models with high sensitivity simulate LGM temperatures around the snowball Earth inception temperature of around 0°C and are usually unstable. This in fact is to be
expected if the inception temperature is controlled by the temperature-dependent behaviour of the climate feedbacks, where the threshold before transiting to a snowball Earth state would be similar across models. Variations of that threshold could be due to for example different parameterizations of sea-ice albedo; tropical shallow clouds, despite considerably increasing the sensitivity of the climate to enter snowball Earth, these feedbacks do not seem to modify the inception temperature (Fig. 3).

It is therefore possible to derive the upper limit of climate sensitivity from the models with stable LGM states (Section 2),
using as snowball temperature threshold a global mean temperature close to 0°C, since both LGM and PI states transit towards a snowball Earth state at similar temperatures (Appendix Fig. A1). In this way, we find that it is implausible that climate sensitivity exceeds 5.5 K (4.4 − 6.6, 90% confidence interval, Fig. 6). This upper limit is close to the sensitivity of CESM2, which simulates an LGM temperature anomaly of −11.3 K (−7.0 K in SSTs) (Zhu et al., 2021b), just above the global mean



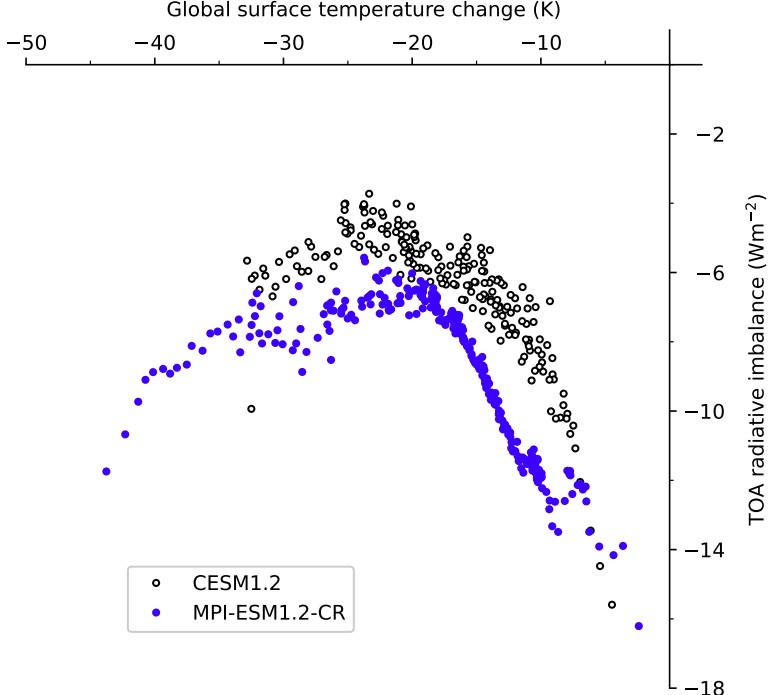

**Figure 5.** Top-of-atmosphere radiative imbalance (Wm$^{-2}$) versus global surface temperature anomaly with pre-industrial conditions (K) in abrupt 1/128xCO2 simulations with the models MPI-ESM1.2-CR and CESM1.2.

temperature of 0°C. Additional support for this limit is obtained from a slightly more sensitive version of the model which is

unstable when run with LGM boundary conditions (Zhu et al., 2022).

This approach on climate sensitivity could have a great potential, in particular as independent constraints from paleoclimates are increasingly useful (Sherwood et al., 2020). Nevertheless, to support this constraint requires an effort from several modelling centres to both publish their LGM simulations, even if they fail to stabilise, and narrow down the uncertainty on the instability threshold. It is important to know whether many models have a threshold gravitating around 0°C, similarly to

MPI-ESM1.2-CR, MPI-ESM1.2-LR and CESM1.2.

To tackle this problem, we offer the following experimental design, which is short and easy-to-replicate:

1. Run a pre-industrial state, with the $CO_2$ concentration set to 1/128 times the PI value, until the model reaches at least the initial instability leading to the snowball Earth state. The chosen $CO_2$ concentration should be a fair trade off between a minimal simulation length and the time-dependency effects discussed in this study. If time allows, complement with less

strong reductions to investigate the time-dependency.





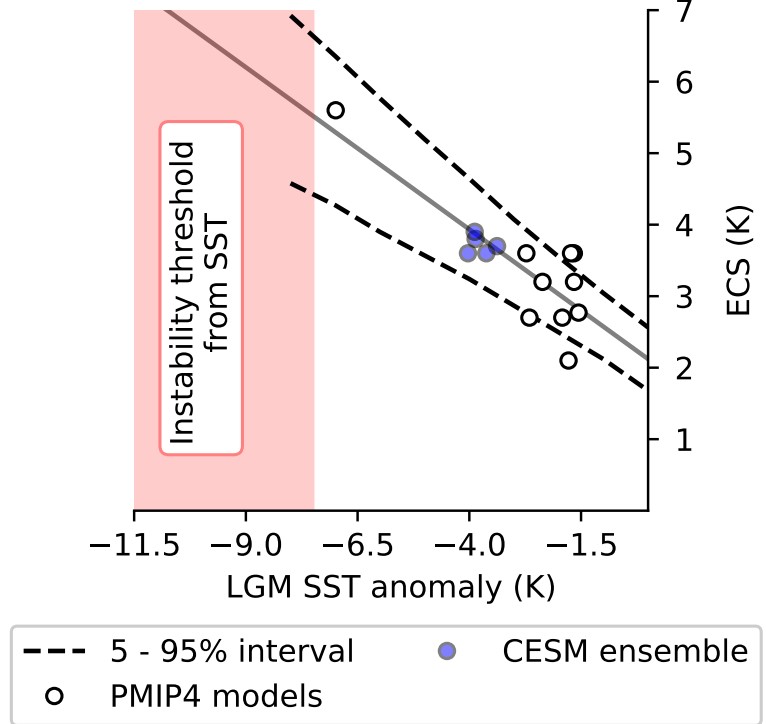

**Figure 6.** Relationship between modelled LGM SST anomaly relative to pre-industrial and the climate sensitivity of PMIP4 models. The relationship lacks robustness, but becomes better when expanded with an ensemble of CESM models simulating the SSTs of the LGM, as seen in Renoult et al. (2023). The critical upper climate sensitivity can be constrained using the temperature at which the climate starts transiting to snowball Earth, which is around -7.5 K from SST anomaly (0°C global mean surface temperature).

2. Generate a plot of top-of-atmosphere radiation imbalance versus surfce temperature anomaly relative to PI, as in Fig. 3. Report the instability threshold temperature, from surface and sea-surface temperature, as well as the global sea-ice concentration.

3. For modelling centres simulating the LGM, we strongly suggest to publish runs which led to instabilities.

The results could support the novel constraint on climate sensitivity presented in this study, but also help understanding instabilities in LGM simulations, a paleoclimate notoriously difficult to model, in particular for higher climate sensitivity models. We advocate that multi-model comparisons of snowball Earth transitions, as well as single-model ensembles with varying climate sensitivity, would better support the relationship between simulated LGM temperatures and the upper bound on model Earth's climate sensitivity. The main strength of this constraint is that it is independent of paleoclimate reconstructions

and simply relies on the fact that Earth did not undergo a transition to a snowball state 21 000 years ago, or during any



earlier glacial cycles during the Pleistocene, providing a valuable additional line of evidence to those already used in the community (Sherwood et al., 2020; Forster et al., 2021).

*Data availability.* The outputs of the simulations shown in this study can be downloaded from https://doi.org/10.5281/zenodo.8117483

*Author contributions.* The idea of the study is of MR. MR performed all simulations, analyses and figures. The paper was written by MR,
JH, NS and TM. TM put the project together.

*Competing interests.* The authors declare that they have no conflict of interest.

*Acknowledgements.* We thank Christoph Braun, Aiko Voigt and Raymond Pierrehumbert for the scientific discussions which helped advancing this study. We thank Eduardo Carril for designing Figure 4. The computations and data storage were enabled by resources provided by the National Academic Infrastructure for Supercomputing in Sweden (NAISS) and the Swedish National Infrastructure for Computing (SNIC) at the National Supercomputer Centre at Linköping University, partially funded by the Swedish Research Council through grant agreements no. 2022-06725 and no. 2018-05973. This research has been supported by the European Research Council, H2020 European Research Council (highECS (770765) and CONSTRAIN (820829)).



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





## Appendix A: Snowball state under solar forcing and from LGM conditions

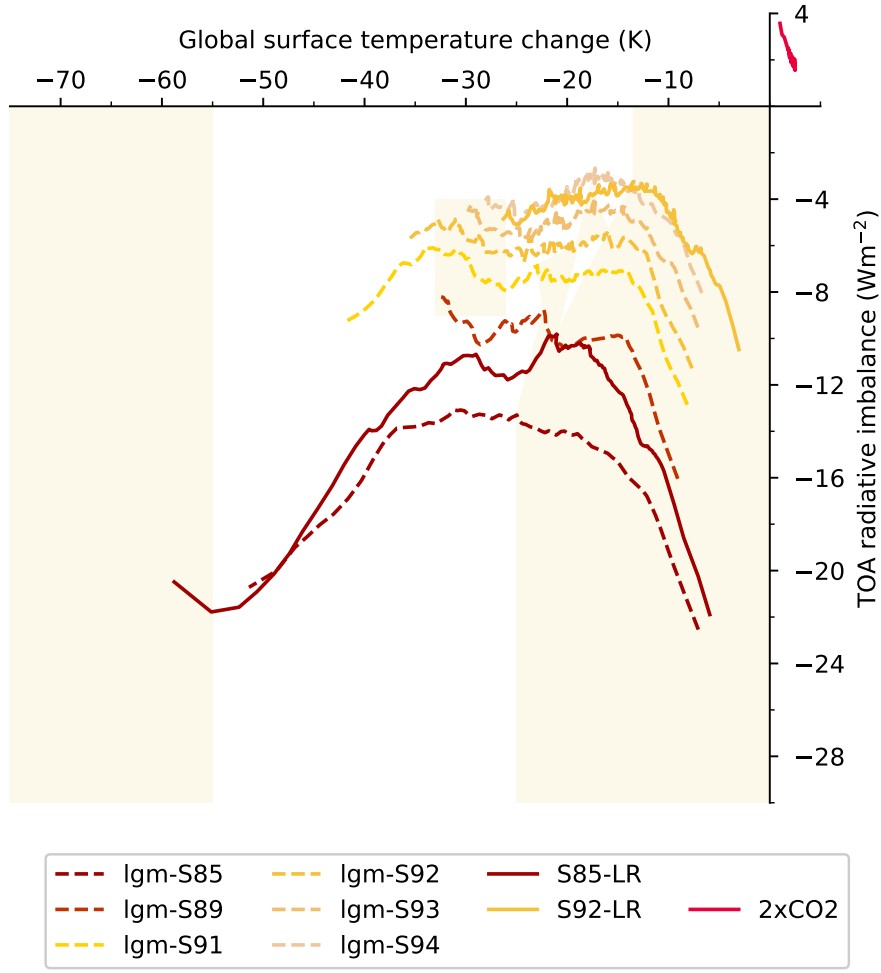

**Figure A1.** Top-of-atmosphere radiative imbalance (Wm$^{-2}$) versus global surface temperature anomaly with pre-industrial conditions (K) under solar forcing from LGM conditions. The similar phases of positive (white) and negative (colored) climate feedbacks as from PI conditions are shown.

We verify how snowball Earth transitions differ when starting from LGM or PI conditions. MPI-ESM1.2-LR is numerically unstable at low $CO_2$ concentrations, so we compare solar-forced snowball Earth transitions between LGM and PI instead.

The LGM is characterized by large ice sheets and lower sea-level which can affect cloud feedbacks as well as global circulations (Muglia and Schmittner, 2015; Sherriff-Tadano et al., 2018; Zhu and Poulsen, 2021; Renoult et al., 2023), thus can influence the transition to a snowball Earth state. Changes in continental distribution and ocean areas are known to affect



inception temperatures. For instance, Voigt et al. (2011) indicates a snowball Earth transition at 96% of pre-industrial solar constant with Marinoan reconstructed paleogeography (∼635 million years ago), which is characterized by agglomerated, bare-soil tropical continents, whereas it would require 94% or lower with PI continents. Tropical land masses reflect more radiations and therefore contribute substantially to snowball Earth transitions.

When initialising from LGM conditions (Fig. A1), the same main phases of stability and instability are identified than from 310 PI conditions. There are slight differences between both cases, for instance the LGM simulation show a single instability phase, whereas the runs starting from pre-industrial still display hints of multiple phases when the solar constant is set at 85% of PI value. However, the inception temperature connected to the main instability leading to the snowball Earth state happens at a broadly similar temperature than from PI conditions, between −15 K and −20 K relative to pre-industrial, which demonstrate the role of the feedback dependency on temperature, irrelevant of the differences in boundary conditions.

Memory effects are expected to differ from LGM and PI conditions and likely explain the slight differences in transition temperatures towards the snowball state. Indeed, the LGM ocean is colder than in PI, and the initial thermal state of the ocean is known to affect the speed at which the snowball state is reached (Bendtsen and Bjerrum, 2002). From the LGM, reaching snowball Earth is therefore easier, as tropical oceans are colder and the evacuation of heat via ocean mixing is enhanced. This should add further in rendering the LGM simulations difficult for modern climate models, as they usually initialise their 320 experiments from LGM states of previous generations.

**Appendix B:  Additional figures**





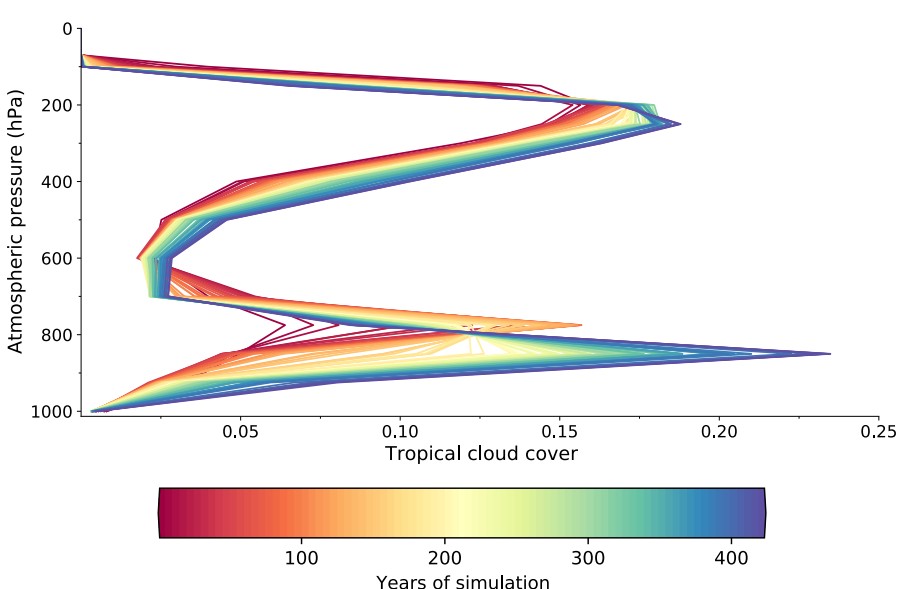

**Figure B1.** Evolution of tropical (30°s - 30°N) cloud cover within the first 15 K of cooling (425 years of simulation) in the 1/16xCO2 simulation.





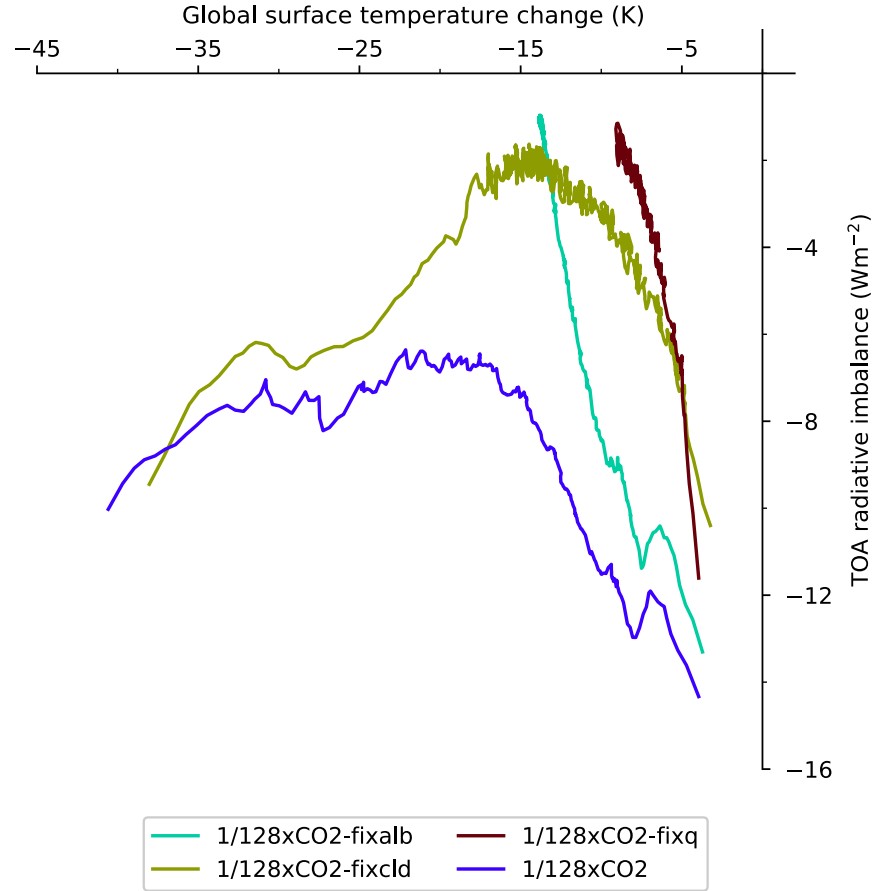

**Figure B2.** Top-of-atmosphere radiative imbalance ($Wm^{-2}$) versus global surface temperature anomaly with pre-industrial conditions (K) for the 1/128xCO2 simulation with either surface albedo (fixalb), water vapor (fixq) or cloud (fixcld) feedbacks locked to that of the control state.





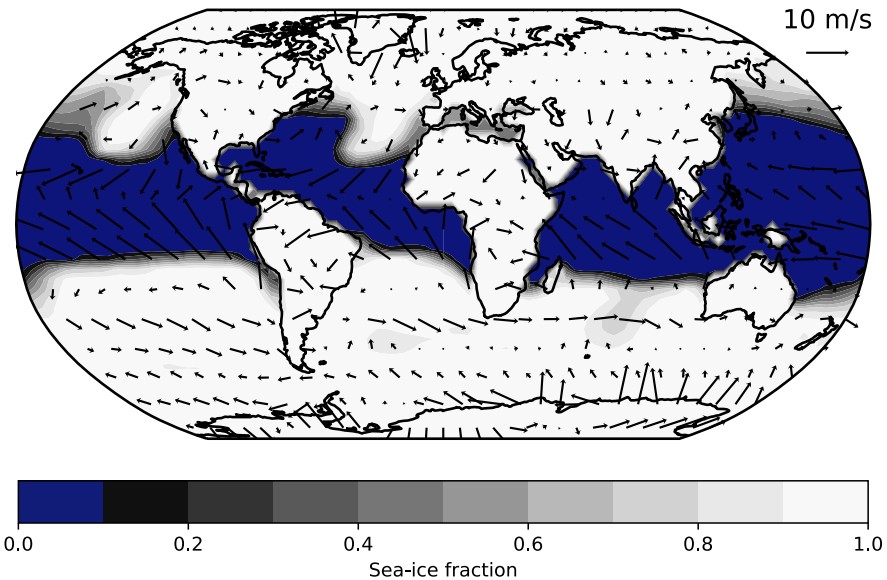

**Figure B3.** Surface wind field on sea-ice cover during the snowball Earth transition for the 1/16xCO2 simulation. Strong winds push sea-ice equatorwards in the tropical regions.

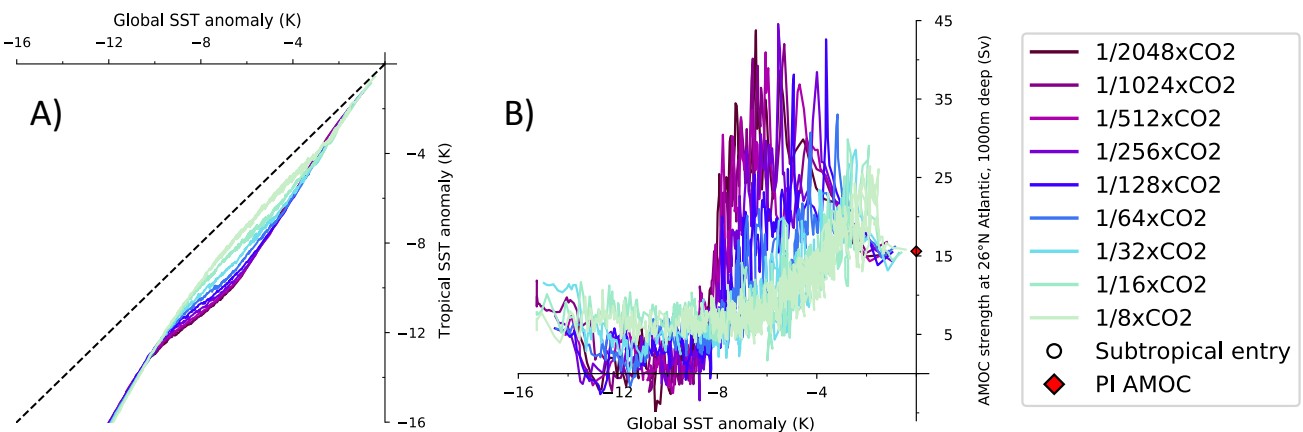

**Figure B4.** Analysis of tropical and global components affecting the rate of snowball Earth transitions. A) Differences in tropical and global SST cooling under all $CO_2$ forcing. B) Strength of the Atlantic meridional overturning circulation (AMOC) measured at 26°N, 1000 m deep (Sv) under all $CO_2$ forcing. The average pre-industrial AMOC strength, around 16 Sv, is highlighted.