# Peer review of "Snowball Earth transitions from Last Glacial Maximum conditions provide an independent upper limit on Earth's climate sensitivity"

_EGUsphere, 2024_

## Author Comment (AC1)

We thank the reviewer for their insightful comments and constructive feedback. Below, we indicate the reviewer's comments in blue and our response in black.

R: The analysis of what feedbacks are important in the snowball Earth transition is not something completely new. For example, the contribution of Pierrehumbert et al. (2011) in the "Snowmip" activity is not mentioned at all (which is odd considering that the authors write about scientifc discussions with Raymond Pierrehumbert and other involved persons in their acknowledgements). Pierrehumbert et al. (2011) specifically discuss that snow and sea-ice albedos together with clouds and atmospheric circulation have a major control of the models behaviour during snowball Earth initiation. There is a short introduction to the topic in the very first paragraph, but I would strongly suggest to add more discussion of the existing literature here.

We added some discussion and references in the Introduction. It is however important to note that Pierrehumbert et al. (2011) summarises past research, notably on Snowball Earth modelling, such as the research of Voigt and Marotzke (2010) or Voigt et al. (2011) which are cited in our paper. Snowmip consists of two coupled models, one of them being ECHAM5/MPIOM which is a previous version of the model we are using and that has been extensively described in papers we cite.

R: l. 31: After the introduction, it is still not clear how the logic behind the ECS constraint works. I would suggest to elaborate on the approach, i.e. that you constrain ECS ultimately by fitting a line for the relationship between a models ECS and the simulated LGM temperature anomaly, and then assume that there is a fixed value of the anomaly at which a snowball Earth would be initiated (in all models). Since a snowball Earth did not happen, you have a new and independent limit on ECS. This reasoning is completely missing in the introduction, so the reader has to guess how this should work. I would suggest to add a bit more explanation in the abstract too, since this is the main point of the paper.

We added some explanation in the Introduction.

R: l. 47-50: How is the growing sea ice problem then treated in MPI-ESM? The reasoning behind having the limit is to avoid numerical artefacts or model crashes as the surface layer runs dry. After reading the rest of the manuscript, it seems like this is just ignored and only the 50 ppm-simulation crashes after 1848 years because of this problem. But why is this not a problem earlier and in the other simulations? This limiter was included, because even pre-industrial states sometimes got too thick ice and crashed. Why are your much colder simulations not crashing quickly?

The sea-ice is left free to grow, until the point where the model stops working, which we refer to as as numerical instability. This is explained in our Methods section: "The growth of thick sea-ice leads to numerical instabilities in the model (> 12 meters). We do not artificially limit sea-ice growth as in other studies (Voigt and Marotzke, 2010; Voigt et al., 2011; Voigt and Abbot, 2012), as it generates latent

heat at the base of sea-ice (Marotzke and Botzet, 2007) which changes the required CO2 forcing for snowball Earth initiation (Hörner et al., 2022)."

Limiting sea-ice thickness can impact the simulation as written above, and is irrelevant for our question as the snowball Earth instability will usually happen before the model reaches numerical instability. The abrupt50ppm run was stopped manually to save resources. It is the only run that does not reach instability; the other runs do crash because of a known behaviour of too much sea-ice in narrow basins, such as the Baltic Sea. This was written only in the caption of Table 1 so we added it in the Methods section as well:

"∗∗The runs were manually stopped and are expected to reach equilibrium in a cold non-snowball state. (Abrupt50ppm)"

R: Methods generally: The authors refer to other papers for their methods, but I think some more explanation should enter also here, to let the reader understand what is being done without having to read up in other papers.

We added a subsection in the Methods which describe the Gregory method (Gregory et al., 2004), and how to read the Gregory plots which are in our paper (Fig.3 and 5)

R: Table 1: It does not really become clear why all the different runs are being done. A bit more thourough explanation of why the individual sets of simulations are being done would make it easier to understand from the beginning, and not after reading the whole manuscrpt.

CO2 has a quasi-logarithmic forcing behaviour so all runs correspond to one half of the previous CO2 concentration, yielding a nearly linear change in radiative forcing. This is a typical approach in climate sensitivity and feedback studies as it simplifies the computation of climate sensitivity. This also allows an easy comparison to simulations at 0.5x, 2x, 4x and 8x pre-industrial CO2 concentrations which are part of CMIP and are often shown in the IPCC reports. We added a few clarifications in the Methods section.

R: When first looking at Fig. 1, I assumed all dots correspond to stable climate states, but later it became clear that these were taken out of transient simulations that are in the middle of approaching a snowball Earth. Over which time periods where they averaged and should these numbers really be produced from non-equlibrium climate states? There generally needs to be more explanation which goes together with the fact that the method section is quite short. There is a similar issue with Fig. 2. These maps are all from one simulation, so over which time periods were they created? A figure with some simple time series of the runs would be very helpful.

Each dot is a one-year average of the same transient simulation as the climate cools down and the top-of-atmosphere energetic imbalance is being reduced. This is a standard approach in the Gregory method as described in Gregory et al. (2004). As this question was common among the reviewers, we added a sub-section which describes the Gregory method and how to read Gregory plots of

Fig.3 and 5, as well was a schematic explaining the approach and how it compares to the real climate system. If the forcing is abruptly set at the beginning of a run, the Gregory plot already acts as a time series: the beginning will always be the right-most point, when temperature is the closest to pre-industrial and the TOA imbalance is equal to the forcing imposed, and the end of the simulation is the left-most point, where in our case the temperature is the coldest. For the maps, they are averaged over a non-fixed time period, as our feedback calculations are averaged over bins of data points spanning 5 degrees of cooling: this is explained for Fig.1 but this was not properly acknowledged for the map. Because the simulations are relatively long, but all of the different length, bins of 5 degrees contain enough years to filter out random patterns and allow comparisons between runs.

R: l. 106-108: First, it is said that positive cloud feedbacks at the sea-ice edge decrease the temperature of instability, then it is said that cloud feedbacks substantially increase the threshold CO2 level, which are opposing statements. I can imagine this dicrepancy comes from comparing local to global effects of the cloud feedback or because neglecting the cloud feedback would decrease global mean temperatures at a reference CO2 level, but please explain in more detail what is the cause of the discrepancy here.

Cloud feedbacks are even more positive at stronger forcing in the initial degrees of cooling (Fig.1), which influences the cooling rate of tropical oceans and affect the temperature of stability. However, interactive cloud feedbacks also allow snowball Earth states to be triggered at much higher CO2 level than when cloud feedbacks are locked (zero). Those two statements are not contradictory as they both refer to local and global effects, as well as the influence of clouds and time and state. We acknowledge the confusion and lack of explanation of Fig.1 and we added details in the text.

R: Fig. 3 description: Terming some phrases "stable climate" even though some of the runs are in the middle of a transition towards a snowball Earth seems incorrect to me. Please elaborate and be more specific about what is actually meant here.

We removed stable/unstable and only kept positive/negative feedback phases.

R: l. 115: To me Fig. 3 does not really show that the different runs transit towards a snowball Earth at roughly the same (transient) global mean temperature. Again, a simple figure with time series of global mean temperature in all the runs would be helpful to support these kind of statements.

We added a subsection on the Gregory method and Gregory plots. A figure with a time series would show exactly the same global mean temperature, which is now shown by the dashed line in the schematic figure we added in the Methods section.

R: Fig. 4: Generally, schematics are nice, but this one is hardly telling a story. Could it be a bit more elaborate or just left out? If you want to keep this in, I am not going to oppose.

We would like to keep it.

R: Chapter 4: I am having a hard time believing that the temperature at which the climate transits to a snowball Earth state is similar across CO2 forcings, setups and even supposedly climate models (l. 118). First of all, what temperature is even meant here? These are all transient simulations, so I assume it is not the global mean temperature of the last stable pre-snowball equilibrium climate (which would surely be highly dependent on the climate model). Is it the (global mean) temperature at which the TOA radiative imbalance starts to grow over time again, as marked by the different phases in Figs. 3 and A1? But then, this is all but a clearly defined temperature range, as can be seen from the large ranges where the imbalance goes sideways in some runs, and even if taking the shown different colors as markers, there is still a range of ~10 K between the different runs. Does this count as "broadly similar"? Especially the statement that this temperature will still be the same even with other climate models seems dubios to me. Again, some of this issue could potentially be resolved by simply explaining more thourougly the procedure that was taken.

We added explanation on the transition temperature in the subsection regarding the Gregory method. The snowball Earth temperature is the point where the TOA imbalance is the least negative: any other year is more unstable than this point, therefore it is the last moment where a stable climate state can exist before transiting to a snowball Earth state. The reason why it is not a fixed value but a range of temperatures is explained in Section 4: "Nevertheless, the transition temperatures of each phase show a slight shift to lower values under stronger negative forcing. Therefore, the climate system deviates from pure state-dependent behaviour as the strength of the radiative cooling and the speed of transition to snowball Earth increases." We also explain a bit further than fast simulations, which are used here to highlight those effects, are not preferable: ". All in all, we suggest slow, low forcing simulations are preferable when analysing snowball Earth transitions, as 1) fast transitions to snowball Earth are hardly realistic, as geological snowball states may form over millions of years (e.g. Schrag et al., 2002) and 2) fast transitions involves temporal effects which would depart from state-dependency".

R: l. 127: Now it sounds like the "inception temperature" is actually the global mean temperature of the last stable pre-snowball climate state? A clear definition of what you mean by this temperature is highly desirable. Also, the final global mean temperature of the 50 ppm run (assuming it reached climatic equilibrium and that a further lowering of CO2 would drive the climate into a snowball state) IS probably the "inception temperature", when defined as above. I don't think that the other transient simulations can give any more reliable input. Hence, simply finding the last stable pre-snowball climate by iteratively changing the CO2 concentration

of individual runs would give a more reliable and more precise estimate of the "inception temperature" for a given model setup.

We added details in the Gregory method subsection. Iteratively changing CO2 concentrations to identify the transition temperature is indeed what we did in this paper. As we point out here: "When abruptly decreasing the CO2 concentration to 50 ppm (around 1/4 of pre-industrial CO2), we find hints of instability near the global mean temperature of 0°C", we indeed believe the inception temperature is close to 0°C. This simulation would be however more than 2000 year long, so continuously iterating around that value would be extremely expensive, and also if our study only focused on values around 50 ppm we could not have been able to discuss state and time dependency over large ranges of CO2 concentrations and thereby relate to other studies.

R: l. 142: I find it very problematic to make such statements. First of all, what is the uncertainty range here? It seems to be in the order of at least 5 K. Second, it should be made clear that if this would be a sound statement, it would only be valid for the modern arrangement of continents and not for the setup during the Neoproterozoic, where the snowball Earth actually occurred. This needs to be specified. Lastly, this temperature is in fact highly model dependent. From personal experience, I can say that parameterisations like the sea-ice and snow albedos or even small changes in parameters of sea-ice dynamics can shift the transition towards snowball Earth inception substantially.

We added the effect of uncertainty on Fig.6, where we show the impact on the constraint is actually minimal compared to previous estimates of the upper bound on ECS. Whether this argument is not valid for Neoproterozoic continents is irrelevant for our question: we actually need to have modern continents to constrain the upper bound of ECS from LGM simulations, as these use modern continents, and the message of this paper is not about the geological past of Earth. While the temperature response to a given forcing could be model-dependent, we believe state-dependency around the inception point could be similar across models, and published studies, including this one, show that it is indeed the case for MPI-ESM1.2, CESM1.2, and CESM2 (Eisenman and Armour, 2024)

l. 151: how does Fig. 5 show that the instability is around 0°C?

We added details on how to read a Gregory plot in the Methods section. Fig.5 shows that CESM1.2 has a similar behaviour and transits at almost the same temperature as MPI-ESM1.2: in the case of a simulation at around 2 ppm of CO2, this is around -5°C. Since we show in Fig.3 that the time-dependency effects move the transition temperature from 0°C to -5°C and below, we expect CESM1.2 to also transits at 0°C under much lower forcing, particularly since it is the case for CESM2 (Eisenman and Armour, 2024). Unfortunately, such simulation would take months to run for CESM1.2 so we decided to save time and resources when running CESM1.2

l. 162: How are the critical ECS and the confidence interval computed? It seems like the values just come from the fit of the regression line in Fig. 6. But how does this account for the uncertainty in the "inception temperature", which surely has an uncertainty range of several degrees, maybe up to 5 K or more. This would substantially increase the uncertainty in the calculated upper limit of ECS.

We added the effect of uncertainty in Fig.6. Even with a large uncertainty of 5 K on the transition temperature, this affects the upper bound of ECS estimate uncertainty by a bit less than 2 K. We emphasise that we are constraining the upper bound of ECS here, not the best estimate value, which is necessarily lower. Much larger values on the upper bound are often given, and with some lines of evidence the upper bound is basically infinite. Therefore, even providing a value of 10 K brings valuable information to the community wide effort of constraining ECS.

l. 179: Point 3 is not really part of the recipe from the points above, but rather a general proposal, hence it doesn't really fit into the list.

Adjusted.

We applied corrections based on the following comments. For the case of 50 ppm written as "close to 1/4", this is a choice we made to be consistent with our runs being halving/doubling of CO2 concentrations from pre-industrial values.

- l.15 and other locations: To my knowledge, it should generally be "sea ice" without the hyphen, but then "sea-ice albedo", i.e. including a hyphen when combined with a following noun.
- l. 18 "referred to as"
- l.38 bad punctuation around MPI-ESM1.2
- l. 69-70: Example of weak language, making it hard to follow the text. "... the highest value of the Earth's ECS that does not lead to an unstable LGM state represents an upper limit..."
- l. 93 "snow ball"?
- l. 125: 50 ppm is rather 1/5 to 1/6 and not 1/4 of PI CO2, why not be precise?
- l. 140: "involve"
- l. 150 "as MPI-ESM1.2"
- l. 156-158: bad punctuation or sentence structure
- l.176 "surface"
- l. 184 "...model Earth's climate sensitivity." What does this man?
- l. 290: The doi in the reference does not go to the actual article, but to an eossar link. Please link the actual article.

---

## Author Comment (AC2)

We are grateful to the reviewer for their valuable suggestions that have enhanced this work. Below, we indicate the reviewer's comments in blue and our response in black.

There are a few inconsistencies and a lack of clarity related to the transition temperature, which I summarize here:
1. How is the transition temperature defined? I assume when the feedback becomes zero?

We added clarifications on that aspect in the Methods section. Yes, the transition temperature is when the feedback becomes zero, but more precisely it is also when the TOA imbalance is the least negative, as any other year would be in a more negative imbalance than this one.

2. If my interpretation is correct, how is the value and the uncertainty determined? E.g. looking at Fig. 3A in the 1/64 x CO2 line, I could see why one would argue the transition temperature to be at -20 K, but at the same time -33 K also seems reasonable. Similar reasonable ranges seem to exist for many other simulations.

See comment above.

3. How does this uncertainty affect the uncertainty of the emergent constraint?

We added the effect of the uncertainty on Fig.6. Even with a large uncertainty of 5 K on the transition temperature, this affects the upper bound of ECS estimate uncertainty by a bit less than 2 K. We emphasise that we are constraining the upper bound of ECS here. Much larger values on the upper bound are often given, and with some lines of evidence the upper bound is basically infinite. Therefore, even providing a value of 10 K with uncertainty brings valuable information to the community effort of constraining ECS.

My second set of comments refers to the emergent constraint:
1. Fig. 6: Taking out all the CESM-2 points (the blue ones and, as I read from the text, the upper left white one) leaves no relationship at all. In fact, the authors write that "the relationship lacks robustness" (caption Fig. 6). Considering only the remaining PMIP models, there seems to be no linear relationship at all, which seems to indicate that all the emergent constraint actually comes from CESM2.

Only the left-most point is CESM2, the others belong to the CESM family, as written in the methods section (CESM1.2, CESM1.3). After reviewing the text, we removed "the relationship lacks robustness", as this is not entirely true. In fact, the relationship is robust with higher ECS models than those of the PMIP4 ensemble. This is shown in Renoult et al. (2023): in fact, CESM2 is not necessary for the robustness of the relationship, as CESM1.2 and CESM1.3 suffice (Fig.14 of Renoult et al., 2023).

2. The instability threshold was previously argued to be around 0 degC, in this figure it starts around -8 K SST anomaly relative to preindustrial. Given that pre-industrial SST were around 15 degC or so, how does this go together?

We agree that this figure is indeed confusing. We used SSTs here because we did not have land surface temperature values for the CESM model family (blue dots), so we instead converted the results we got from Fig.3 into SSTs (which would provide a transition temperature at around -8 K SST relative to pre-industrial). We decided to entirely remove the CESM model family, as described above and in Renoult et al. (2023) they are in fact not necessary for the robustness of the relationship: they and CESM2 are almost interchangeable. Instead, we made a new figure which uses this time only TS to be consistent with Fig.3 and Fig.6 and adjusted our conclusions accordingly.

3. How is the emergent constraint different from simply excluding models that freeze over in the LGM when estimating ECS on the grounds that they would be too sensitive? Would that method lead to the same results?

This is indeed a very good point, however this would only work efficiently if there were enough models with high ECS that would publish their runs. We have very limited information regarding higher ECS models that went to snowball Earth states when trying to simulate the LGM, as explained in the introduction. This is why we try to motivate modelling centres to share and publish those simulations. Using a method similar to an emergent constraint allows to use all the published LGM runs, not just those that fail. Regardless, if we look at the ECS of the models we suspect might be entering snowball Earth instability under LGM conditions, then our constraint appears somewhat conservative. As this is anecdotal evidence, it is difficult to implement in the method..

The third set of comments refers to the threshold of snowball transition at 0 degC:
1. Some arguments are very hand-wavy. For example, the argument that this supposedly generalizes across models (l. 118-119) and the "geometric argument" in l.122-124.

The fact that the state-dependency could be general across models is hypothetical ,but not hand-wavy: sea-ice is expected to freeze at the same temperatures in any model and consistently with our physical understanding, so we believe state-dependency, which is the controlling factor of a true inception towards a snowball state, is similar across models. The geometric argument is admittedly slightly hand-wavy, but not necessarily wrong either: it is a simplification and illustration of the point above.

2. In particular, I am not sure that the transition temperature would be 0 degC across all models. Already in Fig. 3A) I can see a transition temperature range of ~ 15 K across simulations performed with the same model. Similarly, in Fig. 5, the transition temperature seems to be at or below -20 K anomaly to pre-industrial, which would be well below 0 degC, too. The statement that all

models share a similar transition temperature to snowball Earth also seems to be at odds with the statement in l. 28-29, which points towards different models having different transition temperatures. Also the statement in l. 147-148 points toward different transition types and temperatures even within the CESM model family.

The transition temperatures vary a lot in Fig.3 due to time-dependency effects, as written here: "Nevertheless, the transition temperatures of each phase show a slight shift to lower values under stronger negative forcing. Therefore, the climate system deviates from pure state-dependent behaviour as the strength of the radiative cooling and the speed of transition to snowball Earth increases.". However, we also emphasise that slower simulations are the closest to any real transitions, as illustrated in a schematic we added in the Methods section, and because they are the least affected by time-dependent effects. When running a 50 ppm run: "When abruptly decreasing the $CO_2$ concentration to 50 ppm (around 1/4 of pre-industrial $CO_2$), we find hints of instability near the global mean temperature of 0°C". Unfortunately those simulations can run for thousands of years, which is why many studies have used "fast" transitions with strong abrupt forcing, like we also show in our paper. In this case it is then interesting to compare those simulations and discuss time-dependency as we did in our study. L28-29: "These climate models start transiting to a snowball state at temperatures substantially cooler than indicated by LGM reconstructions" does not indicate that models have different transition temperatures: 0°C, which is the transition temperature as calculated in our study, is well below any LGM reconstructions. Some models might have different transition types, for instance they might show a stable waterbelt state. However this does not mean they necessarily would have a different transition temperature. This only indicates that they have a third possible state which exists between today and a complete snowball Earth state.

l. 28 – 29: I found this surprising, and from skimming through Zhu et al. 2021a, I didn't find that information. From my understanding, they look at only one climate model family (CESM), albeit in different configurations. While CESM2 definitely goes to a snowball state, I can't see at what temperature the transition would happen, as their Gregory plot (Fig. 2 (d)) seems to indicate a stable regime throughout (with small, but nevertheless negative feedback). Furthermore, I was under the impression that cloud feedback accelerates the transition to the snowball state, or, as stated in l.95-97, doesn't change the transition temperature. All of these statements seem to be inconsistent with each other. A similar statement is found in l.107. How do these conflicting statements go together?

CESM2 enters runaway below its simulated LGM temperature (so somewhere below -11.3 K). Our sentence is indeed confusing and we modified it. Cloud feedbacks accelerates the transition, but this is not inconsistent with them affecting the transition temperature. Cloud feedbacks can affect the cooling rate of the simulation (how fast the simulation will reach the transition temperature), but the transition will still happen at nearly the same temperature because it is mainly controlled by the strengthening of the sea-ice albedo feedback, as shown in Fig.2 and in the text.

l. 32: independent from what? If the intended meaning is "independent from models", then I think independence is a strong claim, which should be further justified

Independent from other estimates, clarified.

l. 99-100: This is almost exactly the same finding as in Abbot 2014 (https://doi.org/10.1175/JCLI-D-13-00738.1), please cite

Done.

l. 145: "universal": I suggest rewording, given that it was tested on only two selected models

"Universal" is used here in a hypothetical way.

Why is there no summary, conclusions, or discussions section? I don't want to insist on the traditional structure, but some wrap-up and putting-into-context at the end of the paper might be helpful.

We discuss our results within the text and Section 5. We believe the structure of our paper clear is enough to answer our scientific question.

The authors did a great job motivating the study in the introduction. A few additional sentences about ECS and its uncertainty would be helpful, since the emergent constraint on ECS is one of the main purposes of the study. Also, the emergent constraint could already be clearly set as a goal for the paper, as well as the logic that is behind it.

Added.

l.99 I suggest "locked clouds" rather than "locked cloud feedbacks". I find "locked cloud feedbacks" not wrong but misleading, because with locked clouds there is actually zero cloud feedback.

We do not think the term is misleading, as the method applied here is called "feedback locking".

l. 130: "state" here refers to temperature, not $CO_2$ concentration, I guess? If so, then I suggest making this clearer, since the $CO_2$ concentration technically belongs to the state.

To avoid confusions we changed "state-dependency" to "temperature-dependency".

l.156-158: I am not sure the sentence is correct grammatically, at least it's hard to digest

Fixed

l. 171 and following: I suggest to implement the call for the new experiments to the abstract.

Added

---

## Author Comment (AC3)

We appreciate the reviewer's thorough evaluation and recommendations. Below, we indicate the reviewer's comments in blue and our response in black.

L15 The authors state "During the Neoproterozoic (>635 million years ago) …", here the Neoproterozoic ran from ca. 1000 - 538 Mya, and the Cryogenian stage (encompassed the two episodes of the Snowball Earth) ran from ca. 720 – 635 Mya. Your phrasing just needs to be tightened as this would get picked up by someone working in the Precambrian.

To emphasise that our paper is only about system dynamics and not specific events in the geological past of Earth, we removed the Neoproterozoic statement and only refers to "Earth history" in the introduction.

What defines your snowball Earth State? The definition of Snowball Earth generally falls into two camps, the Hard Snowball Earth in which the ocean is fully glaciated with marine terminating tropical glaciers, and a Soft Snowball Earth scenario in which sea ice reached into and perhaps beyond the sub-tropics (extending to approx. 10 - 25 degree latitude, sometimes referred to as waterbelt or slushbelt ). Within the manuscript is snowball Earth inception when the tropical ocean has >0% sea ice fraction (as shown in Figure B3). My apologies if I have missed your definition of snowball Earth inception.

In our case the snowball Earth is an entirely frozen world ocean. However, this specification is irrelevant, as the main point of the manuscript is around the moment the Earth enters instability towards that state.

L25-29 If some PMIP4 models do indeed attain a snowball Earth state in the modelling of the LGM, is this more likely due to deficiencies in model parameterisations? If so, are we really deriving insight into the Earth's climate sensitivity? If my thinking here is logical, this needs a brief discussion in the text.

We are unsure what the reviewer means by "deficiencies". The behaviour of the model is anything but unphysical. A model entering a snowball Earth state as a response to a large negative forcing is completely physical, and it was already shown by Budyko (1969) as an expected behaviour of any numerical model which has a reasonable sea-ice albedo feedback. We believe the reviewer might have meant "unprecedented in geological history". Whether some models experience snowball Earth state due to logical behaviour or some deficiencies within PMIP4 is unfortunately unclear because these runs have never been published. This is what we are promoting in our paper. The only published model we know of that has shown signs of a runaway within PMIP4 is CESM2, and that is also a logical behaviour due to its very high ECS and temperature response.

L28 what models are you referring to when you say these models start transitioning to a snowball state (the PMIP4 models that fail to model the LGM?)

In this sentence we refer mainly to CESM2. We know some models failed to contribute to PMIP4 despite attempting to run the LGM (IPSL model, EC-EARTH

model). Those models have high ECS therefore it could also be due to runaway. Unfortunately those runs are often unpublished.

L38-40. You state that you are using an ESM. It would be beneficial to the reader if you briefly described which sub-models are included (e.g., is there an interactive Ice Sheet Model incorporated? If so is it a global model or just GIS & AIS domains) or are you referring to an atmosphere-ocean coupled model. I assume the latter. I would also write a sentence or two on the sea ice model as the physics and parameterisations would be important to a reader looking to understand and compare your modelling results with their own. It is also relevant to discuss the snow-albedo parametrisation within the model (e.g., deep snow albedo), I am assuming in these simulations that year-round persistent terrestrial snow cover advances equatorward, so a discussion of snow cover within the model is pertinent.

We cite Mauritsen et al. (2019) in our Methods section, which is an extremely detailed description of the model we used. We did not perform any other modification. There is no dynamic ice sheet model. The sea-ice and snow-albedo parametrisation is essentially the same as Voigt and Abbot (2012), except we have melt ponds, which are described in Mauritsen et al. (2019). The model simulates equatorward advancing sea ice, but the snow cover is relatively thin or absent due to the very cold and dry conditions.

L41-43 You state that you use the "coarse" T31 truncated model for PI and the "low resolution" T63 model for LGM. I would be more consistent with your description of resolution, as T63 is higher resolution that T31. Or drop the terms "coarse resolution" and "low resolution" and use "low resolution" and "higher resolution".

Coarse and Low are terms which belong to the nomenclature of the models. MPI-ESM1.2-CR is T31, coarse resolution and MPI-ESM1.2-LR is T63 low resolution. Therefore we will not change it in the text. T63 was standard for the CMIP6 simulations conducted with MPI-ESM1.2 for the latest IPCC report.

Are there any differences in other relevant parameterisations between MPI-ESM1.2-CR (T31 31L) and MPI-ESM1.2-LR (T63 47L) that could impact the outcomes of the study? For example, do both models share the same ocean model (in terms of spatial resolution and sea ice model parameters)?

There are some differences in tuning parameters of the model clouds between the CR and LR models. These are intended to compensate for different model biases caused by the coarse resolution. After reviewing the differences, we have identified that some of the parameter settings are likely to alter slightly the cloud feedbacks. Nevertheless, we note that total feedback and equilibrium climate sensitivity is very similar between CR and LR, and when comparing Fig.3 and Appendix A1, this does not affect our results on a global scale, but could slightly impact local climate feedbacks.

It would be useful for the study if the same run (of Table 1.) was conducted for both CR and LR versions of the model. I understand from Table 1 that some of this is an opportunistic ensemble, and so I understand if this isn't possible in a reasonable time frame.

MPI-ESM1.2-LR is drastically more expensive to run than CR, and that is without including the PRP module which already doubles simulation cost. MPI-ESM1.2-LR is also more sensitive to numerical instabilities and we were not able to run it with the large changes of $CO_2$ concentrations that were used in our study.

L47 A bit unclear here for the reader – are you saying that the act of limiting sea ice growth generates latent heat or does thick sea ice itself generate latent heat?

Sea-ice growth creates latent heat, so if thick sea-ice is being artificially removed then latent heat would be generated in the ocean out of nowhere.

Table 1. 1/2056x $CO_2$ should be 1/2048x $CO_2$?

Fixed.

L56 Why does the length of the run vary with forcing? Could you have run each simulation long enough so that you didn't have to use (linear?) regression of TOA radiation imbalance.

The length of the run is proportional to the initial forcing, as larger forcing will imply a larger cooling rate. All simulations eventually reach a numerical instability due to thick sea-ice, therefore larger forcing will result in shorter runs. It is not possible to run the simulation to a stable snowball Earth state with this version of the model.

L55–57 This is a little bit unclear so needs clarifying- are you computing the climate feedback by determining the change in TOA imbalance over a 5 C temperature change?

Yes, we calculate the slope of TOA imbalance versus surface cooling by taking bins of year spanning 5°C of temperatures. Because the simulations are relatively long, but all of the different length, bins of 5 degrees contain enough years to filter out random patterns and allow comparisons between runs over a relevant range of temperatures.

L59-60 It would be helpful to the reader if you could succinctly describe how these are locked (1-2 sentences) L60 mentions that these are imposed on the radiation parameterization, but this is still not that clear to someone not experienced in locking feedbacks.

This is already described in the Methods section: "These locked-feedback transient simulations read the pre-industrial control albedo, clouds, temperature and humidity and impose them on the radiation parameterization regardless of the changes the system is experiencing, such as the increasing extent of sea-ice." A

large number of other papers use this methodology (e.g. Wetherald and Manage (1988), Mauritsen et al. (2013)). It is not clear what other details the reviewer is expecting.

L64-69 So are you saying that you are comparing the snowball transition state to the modelled LGM state to derive the Equilibrium climate state?

We are using the transition temperature leading to a snowball state as the maximum temperature the Earth can cool down in an LGM state. Since we know a relationship between simulated LGM temperatures, ECS, and we know that during the LGM the Earth did not enter snowball state, we can infer the upper bound on ECS.

L81 be slightly more specific here and say these all relate to global mean surface air temperature.
L91 "southern sea ice edge" I think you mean something like "equatorward sea ice edge" unless you are just considering the northern hemisphere sea ice extent here.

Fixed.

L114 It is not entirely clear from Fig 3. how the climate transition is broadly similar. What feature am I looking at in Figure 3? (is this where TOA radiative imbalance is most +ve?). Do you need to show on Fig 3 the global temperature in which your criteria for snowball Earth is met?

We added an explanation of the Gregory method in the Methods section. The transition is always the point where the imbalance is the least negative, as any other year in the simulation would be in a more negative imbalance than this one.

Figures 3 and A1 have the 2X CO2 in the upper right quadrant extending beyond the numbered parts of the y-axis. I would adjust the x- and y-axis numbering so that they extend into the +ve values.

Fig.3 and A1 have ticks which we believe are helpful enough for a reader to understand the values.

L115 Appendix A should this instead be Figure A1

Appendix A contains Figure A1 and a discussion of that figure.

L125 50 ppm is closer to 1/6 of pre-industrial than ¼

This is a choice we made to be consistent with our runs being halving/doubling of CO2 concentrations from pre-industrial values.

L144 I believe this statement depends upon how tightly you are defining the transition / inception into a snowball Earth state, definition of this transition would clarify this.

We provided a better definition in the methods section.

L161. I am a little uncertain how this 5.5 K (4.4 -6.6, 90% confidence interval) statement relates to the data within Figure 6 – I guess this where your linear fit intercepts with the upper bounds of your instability threshold. Could this be identified within Figure 6.

We added lines on Figure 6 to emphasise that the value and the interval is where our inception temperature crosses with the relationship and its confidence interval.

L301, You state that your model is numerically unstable at low CO2 concentrations and so compare solar-forced snowball Earth transitions instead. Given that you are running at low CO2 levels, has this numerical instability impacted your work?

We focused our work on the transition temperature, which happens at much higher temperatures than the ones close to a numerical instability. Numerical instability due to low temperatures also occur in response to solar forcing.

In the text, be consistent with the term "snow ball" or "snowball"
Some instances of informal scientific language used. 36 "notoriously difficult" or L145 "notoriously performed" are not scientific terms (notorious often means something bad or unfavourable that is somewhat common knowledge). Modify also L46 "few simulations" and L88 "All in all".

After reviewing the text we removed "notoriously" at L36.

Equilibrium climate sensitivity (ECS) is the temperature change to a doubling of CO2 once the atmosphere and ocean (including deep ocean) has equilibrated to that change in energy. I believe here you are in fact just considering only atmospheric changes. I think you therefore need to reconcile this ECS with your definition of climate sensitivity. Here also the definition of your snowball Earth is important (sea ice at the equator? I assume you are holding terrestrial boundary conditions fixed – no expansion of terrestrial glaciation).

No, our ECS follows the same definition as provided in most ECS studies. It includes changes in atmosphere, ocean, vegetation. What is omitted are changes related to methane and other chemical and aerosol feedbacks, however those changes are omitted already in most LGM runs and ECS estimates, therefore the impact can be neglected (Renoult et al., 2023). According to IPCC AR6, the ECS should be defined without changes to ice sheets.